# The Naples Prognostic Score Is a Useful Tool to Assess Surgical Treatment in Non-Small Cell Lung Cancer

**DOI:** 10.3390/diagnostics13243641

**Published:** 2023-12-12

**Authors:** Stefano Elia, Alexandro Patirelis, Georgia Hardavella, Antonella Santone, Federica Carlea, Eugenio Pompeo

**Affiliations:** 1Department of Medicine and Health Sciences, University of Molise, 86100 Campobasso, Italy; antonella.santone@unimol.it; 2Thoracic Surgery Unit, Tor Vergata University Hospital, 00133 Rome, Italy; alexandro.patirelis@hotmail.it (A.P.); pompeo@uniroma2.it (E.P.); 39th Department of Respiratory Medicine, Athens Chest Diseases Hospital Sotiria, 11527 Athens, Greece; georgiahardavella@hotmail.com

**Keywords:** non-small cell lung cancer, prognosis, Naples Prognostic Score, thoracic surgery, survival, prognostic score

## Abstract

Different prognostic scores have been applied to identify patients with non-small cell lung cancer who have a higher probability of poor outcomes. In this study, we evaluated whether the Naples Prognostic Score, a novel index that considers both inflammatory and nutritional values, was associated with long-term survival. This study presents a retrospective propensity score matching analysis of patients who underwent curative surgery for non-small cell lung cancer from January 2016 to December 2021. The score considered the following four pre-operative parameters: the neutrophil-to-lymphocyte ratio, lymphocyte-to-monocyte ratio, serum albumin, and total cholesterol. The Kaplan–Meier method and Cox regression analysis were performed to evaluate the relationship between the score and disease-free survival, overall survival, and cancer-related survival. A total of 260 patients were selected for the study, though this was reduced to 154 after propensity score matching. Post-propensity Kaplan–Meier analysis showed a significant correlation between the Naples Prognostic Score, overall survival (*p* = 0.018), and cancer-related survival (*p* = 0.007). Multivariate Cox regression analysis further validated the score as an independent prognostic indicator for both types of survival (*p* = 0.007 and *p* = 0.010, respectively). The Naples Prognostic Score proved to be an easily achievable prognostic factor of long-term survival in patients with non-small cell lung cancer after surgical treatment.

## 1. Introduction

Lung cancer is one of the most common malignancies and the main cause of cancer death in men and women combined worldwide [1,2]. Non-small cell lung cancer (NSCLC) accounts for 84% of all lung cancers [3]. Despite the improvements in early lung cancer detection and treatment options, about 30–50% of patients with completely surgically resected lung cancer develop recurrence [4,5,6], and 5-year survival ranges between 40 and 90% [7]. 

Recently, there has been a growing interest in finding possible prognostic markers that might impact management plans. The early identification of patients with a higher probability of a poor outcome can potentially guide early personalized treatment. Currently, many hematological markers, which can be easily obtained in daily clinical practice, are increasingly utilized for the prognosis of several cancers, including NSCLC. In particular, systemic inflammation and nutritional status have been proven to be involved in cancer development [8,9], and related biomarkers have been evaluated as possible indicators of outcomes for oncologic patients [10,11,12,13,14]. 

In 2017, Galizia et al. proposed a novel score, the Naples Prognostic Score (NPS), based on both inflammatory and nutritional biomarkers for patients receiving surgery for colorectal cancer [15]. The score considered the pre-operative neutrophil-to-lymphocyte ratio (NLR), lymphocyte-to-monocyte ratio (LMR), serum albumin, and total cholesterol and was proven to have a strong association with long-term survival. 

In this retrospective study, we evaluated whether NPS is associated with disease recurrence and death in a group of patients with surgically resected NSCLC.

## 2. Materials and Methods

### 2.1. Patient Selection

This study presents a retrospective analysis of patients who underwent surgery for NSCLC confirmed by their final histology from January 2016 to December 2021. All enrolled patients have discussed in a multidisciplinary team (MDT) meeting and subsequently underwent pulmonary resection (pneumonectomy, bilobectomy, lobectomy, segmentectomy, wedge resection) and lymphadenectomy. All patients’ clinical pre-operative nodal and metastatic stage was N0 and M0, and the clinical stage ranged from stage I to stage IIIA. Patients with a history of infection, any surgery within the previous 3 months, or any malignancy within the last 5 years preceding pulmonary resection were excluded. We also excluded anyone who had a positive history of hematological, autoimmune, or immunodeficiency diseases. We finally excluded patients who had incomplete pre-operative laboratory tests. 

### 2.2. Data Collection

Data about patients’ demographic and clinical information were collected from clinical records, including sex, age, comorbidities, smoking history, the type of surgery, final histology, pT, pN, and pre-operative laboratory tests about neutrophils, lymphocytes, monocytes, serum albumin, and total cholesterol. NPS was calculated, as stated by the original study of Galizia et al. [15], considering NLR, LMR, serum albumin, and total cholesterol. According to previous studies, 1 point was assigned if NLR was ≤2.96, LMR was ≥4.44, serum albumin was <4 g/dL or total cholesterol was ≤180 mg/dL. Then, patients were divided into 3 groups according to their final score as follows: group 0 for a final score of 0, group 1 for a final score of 1 or 2, and group 2 for a final score of 3 or 4.

Patient follow-up data were collected from outpatient clinic records, medical inpatient records, or virtual consultations. We measured the disease-free survival (DFS) as the time from the day of surgery to recurrence and the overall survival (OS) as the time from surgery to patients’ deaths. We also evaluated cancer-related survival (CRS), considering only deaths due to NSCLC. 

### 2.3. Objective

The main goal of this retrospective study was to investigate whether NPS is related to long-term survival in operated patients with NSCLC. Particularly, we evaluated if patients within a higher Naples group presented an increased tendency of recurrence or a poorer prognosis. We also assessed if the NPS had a higher prognostic value compared to its single biomarkers.

### 2.4. Statistics

Statistical analysis was performed using SPSS (IBM Corp. Released 2016. IBM SPSS Statistics, Version 26.0, Armonk, NY, USA: IBM Corp.), and a *p*-value less than 0.050 was considered statistically significant. 

Continuous variables were reported as the median and interquartile range (IQR) and categorical ones as whole numbers and percentages.

Receiver operating characteristic (ROC) curves were applied to evaluate the ability of NPS groups, NLR, LMR, serum albumin, and total cholesterol to predict prognosis by comparing their area under the curves (AUCs).

Prognostic factor evaluation was initially based on survival curves using the Kaplan–Meier method and log-rank test for DFS, OS, and CRS. Afterward, univariate Cox regression was performed. The covariates taken into consideration were as follows: age (median, ≤72 vs. >72 years), gender (male vs. female), smoking history (never smoked vs. former or current smoker), surgical procedure (major, including pneumonectomy, bilobectomy and lobectomy vs. sublobar, including both segmentectomy and wedge resection), the side of surgery (right vs. left), the lobe affected by malignancy (upper or middle vs. lower), pT (1 vs. 2, 3 or 4), pN (0 vs. 1 or 2), histology (adenocarcinoma vs. squamous cell carcinoma) and Naples group. Factors significantly affecting survival during univariate analysis underwent multivariate analysis.

After this preliminary analysis, we performed propensity score matching to reduce possible selection bias. The two considered populations were the Naples group 0 and 1 vs. Naples group 2, and they were selected and matched one by one. This division was made according to the results of the Kaplan–Meier analysis, with similar survival rates for groups 0 and 1 after taking into consideration previous studies [16,17,18]. The covariates considered for this model were as follows: age (≤72 vs. >72 years old), gender, smoking history, the type of surgery (major vs. sublobar resection), pT (T1 vs. T2-3-4), pN (N0 vs. N1-2) and histology. To verify the homogeneity between these two groups, the standardized difference was calculated for each covariate before and after matching. Subsequently, we repeated Kaplan–Meier, and Cox regression analysis with the new population. 

## 3. Results

### 3.1. Demographic and Clinical Characteristics

The demographic and clinical features of the enrolled population, consisting of 260 patients, are summarized in Table 1. The median age of patients submitted to surgery was 72 (IQR 65–77), and 64.6% of them were male. Half of the population was a former smoker, while only a minority (37/260, 14.2%) denied any smoking history. The most common type of surgery was lobectomy (187/260, 71.9%), followed by wedge resection (51/260, 19.6%), segmentectomy (10/260, 3.8%), pneumonectomy (8/260, 3.1%) and bilobectomy (4/260, 1.5%). Over 2/3 of patients had chronic obstructive pulmonary disease, 40% had emphysema, and ~10% had a diagnosis of coronary artery disease. Wedge resection was offered to patients with peripheral lesions and borderline lung function due to significant underlying chronic obstructive pulmonary disease with impaired lung function. The final histology showed lung adenocarcinoma in 184 patients (70.8%). Regarding pathological staging, the majority of cases were stage I or II; 115/260 (44.2%) patients were pT1, and 103/260 (39.6%) were pT2, while 212/260 (81.5%) had no regional lymph nodes involved in the pathology review. A Charlson index would have been challenging to resolve due to the study orientation.

With reference to NPS, 86/260 patients (33.1%) had an NLR > 2.96, and 205/260 (78.8%) had an LMR < 4.44. Regarding nutritional markers, serum albumin was < 4g/dL in 99/260 patients (38.1%), and half of them had total cholesterol scores ≤180 mg/dL (127/260, 48.8%). Therefore, 28/260 patients belonged to Naples group 0, 146/260 to group 1 (56/260 with a score of 1 and 90/260 with a score of 2), and 86/260 to group 2 (63/260 with a score of 3 and 23/260 with a score of 4).

### 3.2. Follow-Up

The median follow-up was 26 months (IQR 15–40 months). A total of 93/260 patients (35.8%) presented recurrence during this period, with a median time to recurrence of 16 months (IQR 8–29 months). A total of 44/260 deaths (16.9%) occurred, and more than half of them were due to lung cancer (24/44, 54.5%). The median time to death for these patients was 13 months (IQR 6–22 months).

### 3.3. ROC Curves

NPS was found to have the largest AUC for all the considered outcomes when compared to NLR, LMR, serum albumin, and total cholesterol. In particular, the AUC values were 0.58 (95% CI 0.51–0.66, *p* = 0.025) for the risk of recurrence, 0.67 (95 CI 0.59–0.76, *p* < 0.001) for the risk of death, and 0.71 (95% CI 0.60–0.81, *p* = 0.001) for the risk of cancer-related death. The AUCs for all other variables are reported in Figure 1.

### 3.4. Survival Analysis 

We analyzed the influence of each NPS parameter on survival with Kaplan–Meier curves. There was no statistical significance between the NLR and survival, while patients with LMR < 4.44 had shorter disease-free survival (*p* = 0.031). During the analysis of nutritional markers, both serum albumin and total cholesterol affected OS (*p* = 0.003 and *p* < 0.001, respectively) and CRS (*p* = 0.020 and *p* = 0.004, respectively), while they had no influence on DFS.

After this preliminary analysis, we utilized Kaplan–Meier curves to evaluate the differences among the Naples groups for survival. A statistically significant difference was found among the groups in the DFS (*p* = 0.037), as shown in Figure 2. Recurrence occurred in 6/28 patients in group 0 (21.4%), 48/146 patients in group 1 (32.9%) and 39/86 patients in group 2 (45.3%). Five-year DFS was 55.8% for group 0, 44.4% for group 1 and 37.8% for group 2.

NPS also affected OS, as reported in Figure 3 (*p* < 0.001). We recorded 1 case of death in group 0 (1/28, 3.6%), 17 cases in group 1 (17/146, 11.6%), and 26 cases in group 2 (26/86, 30.2%). Five-year OS was 92.9% for group 0, 83.8% for group 1, and 59.4% for group 2.

Finally, a significant difference was found in the CRS for *p* = 0.001 (Figure 4). Deaths due to cancer were recorded in 8/146 individuals in group 1 (5.5%) and 16/86 individuals in group 2 (18.6%), while no cases were registered in group 0. The five-year CRS was 100.0% for group 0, 91.4% for group 1, and 72.2% for group 2.

Cox regression analysis (Table 2) showed that a higher Naples group was a predictor of shorter DFS only during univariate analysis (*p* = 0.011). Conversely, for multivariate analysis, a higher Naples group was associated with worse OS (HR = 2.5, 95% CI 1.4–4.3, *p* < 0.001) together with a pT > 1 (HR = 3.5, 95% CI 1.5–7.9, *p* = 0.003). Similarly, a decreased CRS was associated with a higher Naples group (HR = 3.5, 95% CI 1.6–7.9, *p* = 0.002), pT > 1 (HR = 4.0, 95% CI 1.2–13.8, *p* = 0.027), and the presence of metastatic regional lymph nodes at final histology (HR = 2.8, 95% CI 1.3–6.3, *p* = 0.015).

### 3.5. Propensity Score Matching

The standardized difference before and after matching for each covariate is reported in Table 3. The propensity score matching extrapolated a final population of 154 patients, 77 of whom belonged to Naples group 0 or 1 and 77 to Naples group 2.

As shown in Figure 5, DFS did not present significant differences between Naples groups 0 and 1 vs. group 2. In detail, Naples groups 0 and 1 had a 5-year DFS rate of 38.0%, and Naples group 2 had 41.2% (*p* = 0.34). On the other hand, a significant difference was confirmed between groups 0-1 and group 2 in OS (*p* = 0.018), with a 5-year survival rate of 83.5% and 61.1%, respectively (Figure 6), and for CRS (*p* = 0.007), 5-year survival rates were identified for 95.3% of Naples group 0 and 1 and 74.7% of Naples group 2 (Figure 7).

The Cox regression analysis (Table 4) showed that NPS did not affect DFS during univariate analysis (*p* = 0.34). Conversely, it resulted in a significant prognostic factor in multivariate analysis for OS (HR = 2.5, 95% CI 1.2–5.2, *p* = 0.018) together with pT (HR = 5.2, 95% CI 1.6–17.0, *p* = 0.007), and the only significant prognostic factor for CRS (HR = 5.2, 95% CI 1.5–18.2, *p* = 0.010). 

## 4. Discussion

Systemic inflammation has been proven to play a key role in tumorigenesis, and several studies demonstrate that some inflammatory biomarkers from routinary laboratory tests could be a predictor of long-term outcomes in patients with NSCLC. For example, neutrophils stimulate angiogenesis by secreting proangiogenic factors and are involved in the production of growth factors. Consequently, neutrophilia is usually related to a poorer prognosis [19]. Monocytes stimulate tumor angiogenesis by producing the vascular endothelial growth factor (VEGF), and they can also differentiate into tumor-associated macrophages, thus favoring the creation of a tumor microenvironment [20,21]. Tumor angiogenesis and growth also seem to be promoted by an increased number of platelets, which release VEGF and whose proliferation is stimulated by pro-inflammatory cytokines [22,23]. On the contrary, lymphocytes react against cancer by inhibiting cellular proliferation and migration, and their high levels may correlate with a positive prognosis [24].

Based on the above, new prognostic tools such as pre-operative NLR, LMR, and the platelet to lymphocyte ratio (PLR) started to be used, achieving a strong correlation with all lung and other types of cancer prognosis [8,25,26].

Systemic inflammation might usually be related to nutritional impairment due to an increase in catabolic processes and energy consumption. Therefore, low levels of nutritional markers such as serum albumin, whose synthesis is inhibited by systemic inflammation [27], or cholesterol, which is a pivotal component of cellular membranes and is involved in cell homeostasis [28], could be considered a bad prognosticator [29,30]. Pre-operative nutritional scores such as the Prognostic Nutritional Index (PNI) or the Controlling Nutritional Status (CONUT) score are independent prognostic factors in lung cancer [31,32]. 

In 2017, Galizia et al. assessed the Naples Prognostic Score, which was a new prognostic tool considering both inflammatory (NLR and LMR) and nutritional biomarkers (serum albumin and total cholesterol) comprehensively [15]. NPS proved to be a prognostic factor for colorectal cancer in terms of OS and DFS. NPS was also applied to pancreatic cancer [18], osteosarcoma [33], endometrial cancer [34], gastric [35], and esophageal cancer [36], showing a significant correlation with DFS [33,34], OS [18,33,34,35] or CRS [36].

To the best of our knowledge, only a few studies have analyzed the role of NPS in NSCLC. Guo et al. evaluated patients with unresectable stage III NSCLC and showed NPS to be an independent prognostic factor for both DFS and OS [37]. Similar results were obtained by Zou et al., who studied NPS in patients with locally advanced NSCLC following neoadjuvant therapy [38]. Both studies included inoperable patients with locally advanced disease. These patients are affected more by cachexia and lower albumin levels at the time of diagnosis in comparison to patients from earlier lung cancer stages, and this impacts any potential clinical validation of NPS in NSCLC [39,40,41,42].

Our study involved a range of operable NSCLC patients and, to the best of our knowledge, is the first one to show a correlation between NPS and CRS in these patients. Early-stage NSCLC patients who underwent an operation were also studied by Dahu Ren et al. [43], who found a strong correlation between NPS with recurrence-free survival (RFS) and OS, while Li et al. [16] found a correlation between DFS and OS. The latter included only patients undergoing thoracoscopic surgery, alluding to a narrower selection of patients included in their study as opposed to our study, which included a wider selection of patients in terms of their operability and surgical approach, therefore representing conditions in a pragmatic clinical setting. Finally, Peng et al. reported NPS as a significant prognosticator for both DFS and OS in all patients with NSCLC [44]. Our study proves that there is a strong association between NPS groups and OS and CRS, as shown using Kaplan–Meier analysis and multivariate Cox regression. These results were confirmed after propensity score matching, which balanced potential confounding factors regarding the clinical and pathological features of the enrolled population between Naples groups 0-1 and 2. Regarding DFS, NPS proved to be a significant indicator of prognosis only in the pre-propensity analysis with the Kaplan–Meier method and univariate Cox regression. Unlike the above studies, we also took into account CRS, which evaluates only deaths due to cancer and is more specific than OS.

Our findings confirm that NPS may be a strong predictor of long-term survival outcomes in patients with NSCLC following surgical resection. Our sample included a 19.6% wedge resection rate due to the patient’s underlying COPD affecting lung function. Although anatomical resection is the golden standard in lung cancer surgery, a tailored yet pragmatic approach is required in patients with early-stage lung cancer and underlying COPD with impaired lung function. Considering that our country has almost 800,000 more smokers than in 2019 and the consumption of heated tobacco products has almost tripled, these patients constitute a non-negligible group in real-life clinical settings that require a tailored surgical approach and a more intense smoking cessation campaign [45]. Although lobectomy remains the gold standard in the surgical treatment of NSCLC, sublobar anatomical and wedge resection may still be considered at the lung cancer MDT and within a team of operating surgeons to ensure the optimal approach is offered to patients.

NPS is simple to use in daily clinical practice as it utilizes parameters that are readily available in patients after undergoing thoracic surgery with curative intent. Following extensive validation, it may help in combination with clinical and radiological aspects to inform the decision-making process with regard to treatment and interventions. 

This study has some limitations. It is a single-center retrospective study, thus making possible selection bias possible. We addressed this by performing propensity score matching, which reduced the differences in clinical and pathological features between the groups. 

Another limitation is the small sample size, which was further reduced after propensity score matching, thus reducing the possibility of generalizing these results. The propensity score did not include comorbidities. As this was a real-life retrospective study, all our patients had significant comorbidities (the median number was 3), and this limited the propensity score matching, therefore impacting the overall assessment and that of NPS. The team considered that an extensive dataset would be required to properly match this, and then broad assumptions could be made; however, these would be difficult to prove, thus limiting the impact of the sample.

Therefore, further larger multicentric studies are needed to validate these findings.

## 5. Conclusions

NPS is an easily obtainable index that comprehensively considers the inflammatory and nutritional status of patients with NSCLC. It was proved to be a significant prognostic factor of long-term survival outcomes in patients with NSCLC after surgical treatment. If validated by further multicentric studies, NPS could be considered as a factor to tailor individualized treatment in patients with a higher risk of poor outcomes.

## Figures and Tables

**Figure 1 diagnostics-13-03641-f001:**
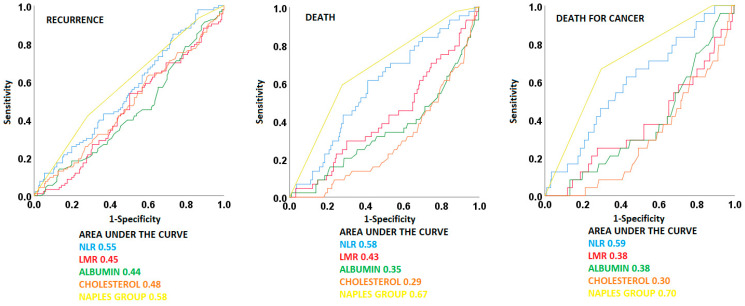
Receiver operating characteristic (ROC) curves for Naples Prognostic Score and its variables.

**Figure 2 diagnostics-13-03641-f002:**
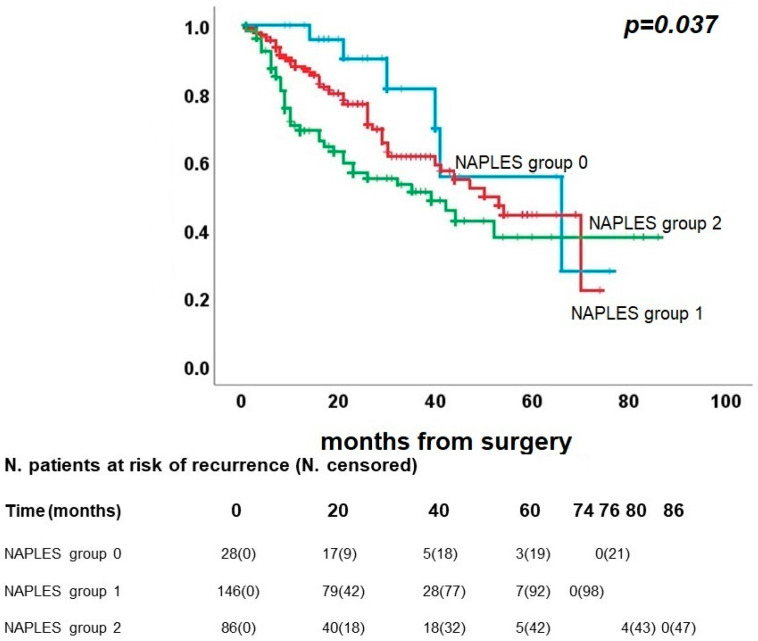
Kaplan–Meier curve for disease-free survival.

**Figure 3 diagnostics-13-03641-f003:**
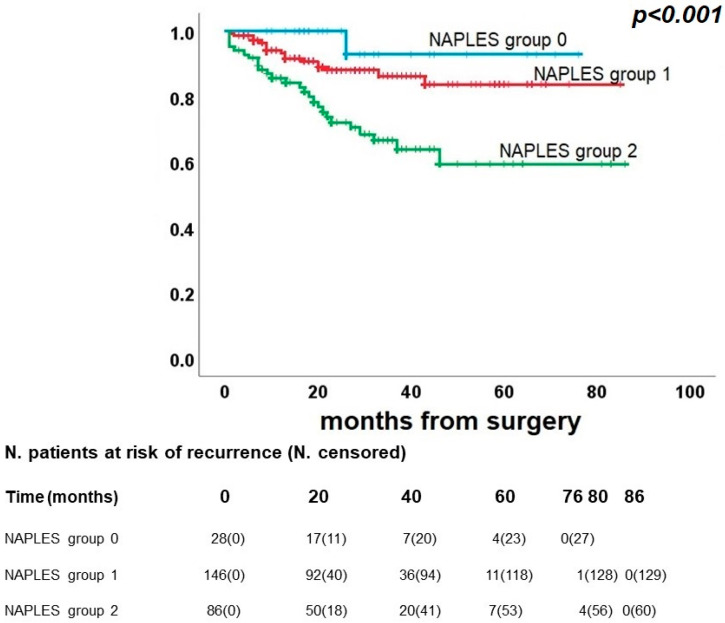
Kaplan–Meier curve for overall survival.

**Figure 4 diagnostics-13-03641-f004:**
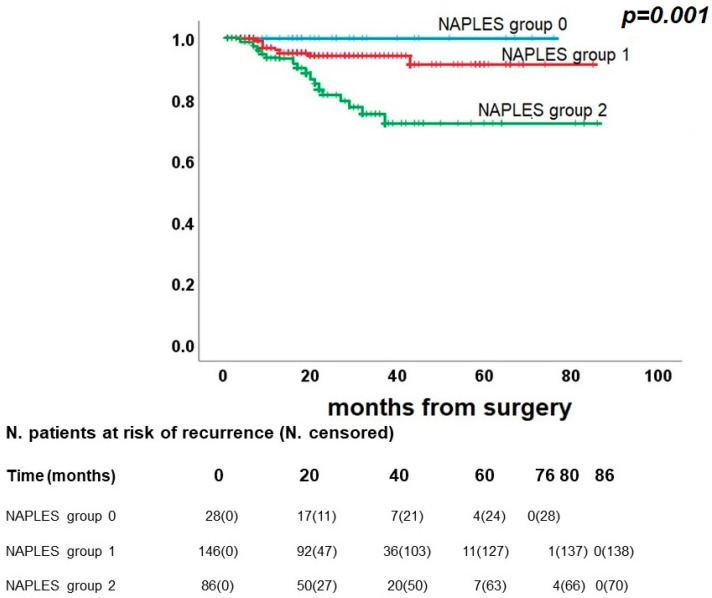
Kaplan–Meier curve for cancer-related survival.

**Figure 5 diagnostics-13-03641-f005:**
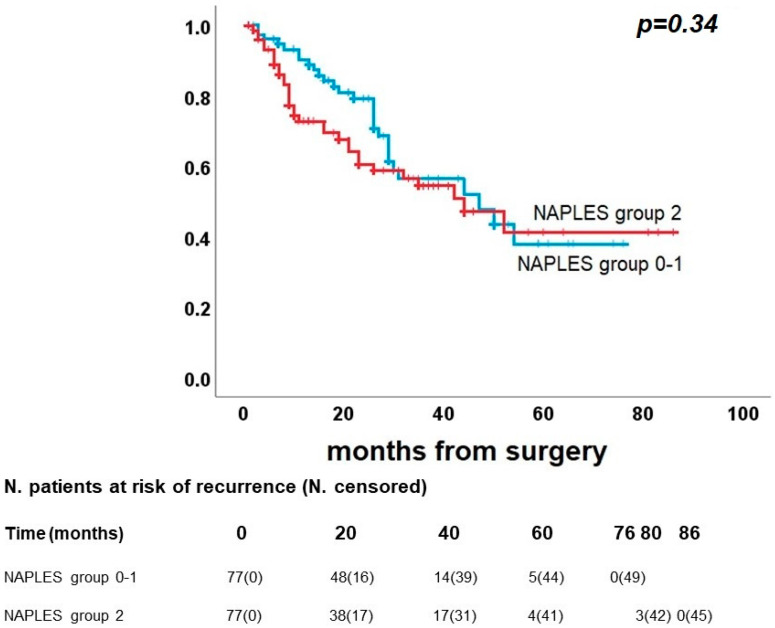
Post-propensity Kaplan–Meier curve for disease-free survival.

**Figure 6 diagnostics-13-03641-f006:**
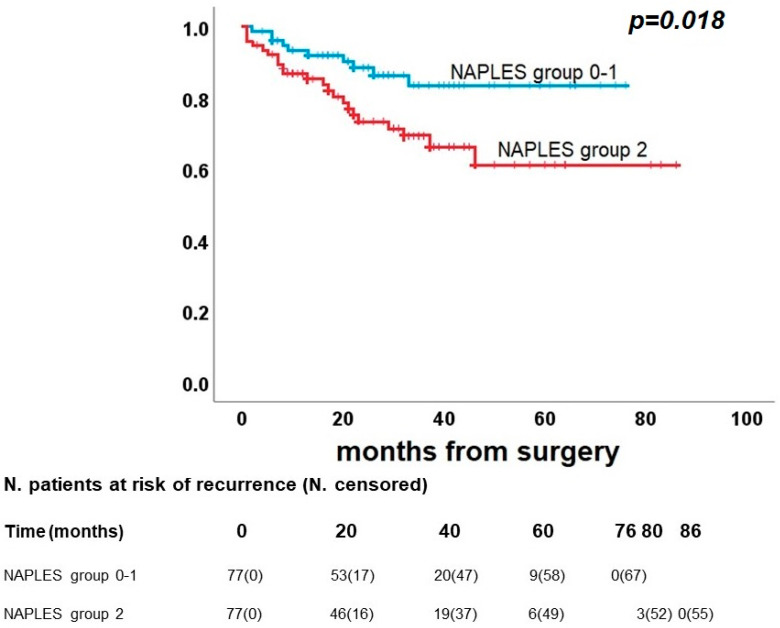
Post-propensity Kaplan–Meier curve for overall survival.

**Figure 7 diagnostics-13-03641-f007:**
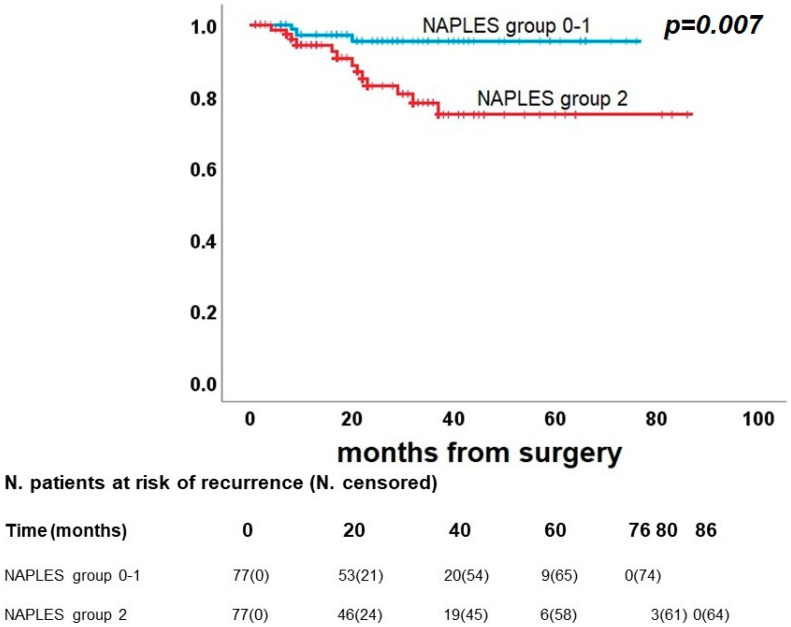
Post-propensity Kaplan–Meier curve for cancer-related survival.

**Table 1 diagnostics-13-03641-t001:** Demographic and clinical characteristics of the enrolled population. IQR: interquartile range.

Variable	
Median age, years (IQR)	72 (65–77)
Gender, *n* (%)	
Male	168 (64.6%)
Female	92 (35.4%)
Median number of comorbidities (IQR)	3 (IQR 2–5)
Smoking history, *n* (%)	
Never smoked	37 (14.2%)
Former smoker	130 (50.0%)
Current smoker	93 (35.8%)
Surgical procedure, *n* (%)	
Pneumonectomy	8 (3.1%)
Bilobectomy	4 (1.5%)
Lobectomy	187 (71.9%)
Segmentectomy	10 (3.8%)
Wedge resection	51 (19.6%)
Surgical access, *n* (%)	
Thoracoscopy	224 (86.2%)
Thoracotomy	36 (13.8%)
Side of surgery, *n* (%)	
Right	151 (58.1%)
Left	109 (41.9%)
Lobe (pneumonectomies excluded), *n* (%)	
Upper	151 (58.1%)
Middle/lingula	11 (4.2%)
Lower	90 (37.7%)
Final histology, *n* (%)	
Lung adenocarcinoma	184 (70.8%)
Lung squamous carcinoma	76 (29.2%)
pT, *n* (%)	
1	115 (44.2%)
2	103 (39.6%)
3	29 (11.2%)
4	13 (5.0%)
pN, *n* (%)	
0	212 (81.5%)
1	29 (11.2%)
2	19 (7.3%)
Neutrophil/lymphocyte ratio, *n* (%)	174 (66.9%)
≤2.96	86 (33.1%)
>2.96	
Lymphocyte/monocyte ratio, *n* (%)	
<4.44	205 (78.8%)
≥4.44	55 (21.2%)
Serum albumin, *n* (%)	
<4.0 g/dL	99 (38.1%)
≥4.0 g/dL	161 (61.9%)
Total cholesterol, *n* (%)	
≤180 mg/dL	127 (48.8%)
>180 mg/dL	133 (51.2%)
NAPLES score, *n* (%)	
0	28 (10.8%)
1	56 (21.5%)
2	90 (34.6%)
3	63 (24.2%)
4	23 (8.8%)
NAPLES group, *n* (%)	
0	28 (10.8%)
1	146 (56.2%)
2	86 (33.1%)
Median follow-up, months (IQR)	26 (15–40)
Recurrence, *n* (%)	
Yes	93 (35.8%)
No	167 (64.2%)
Median time to recurrence, months (IQR)	16 (8–29)
Status, *n* (%)	
Alive	216 (83.1%)
Dead	44 (16.9%)
Cancer-related death, *n* (%)	
Yes	24 (54.5%)
No	20 (45.5%)
Median time to death, months (IQR)	13 (6–22)

**Table 2 diagnostics-13-03641-t002:** Cox regression analysis. CI: confidence interval; F: female; HR: hazard ratio; M: male.

	Disease-Free Survival			Overall Survival			Cancer-Related Survival		
	Univariable	Multivariable		Univariable	Multivariable		Univariable	Multivariable	
	*p*-Value	HR (95% CI)	*p*-Value	*p*-Value	HR (95% CI)	*p*-Value	*p*-Value	HR (95% CI)	*p*-Value
Age (≤72 vs. >72) years	0.019	1.4 (0.9–2.1)	0.14	0.057	-	-	0.11	-	-
Gender (M vs. F)	0.13	-	-	0.10	-	-	0.10	-	-
Smoking history (never vs. former/current)	0.84	-	-	0.080	-	-	0.41	-	-
Surgical procedure (major vs. sublobar)	0.028	1.7 (1.1–2.7)	0.020	0.98	-	-	0.78	-	-
Side of surgery (right vs. left)	0.057	-	-	0.67	-	-	0.51	-	-
Lobe (upper and middle vs. lower)	0.66	-	-	0.33	-	-	0.25	-	-
pT (1 vs. 2-3-4)	<0.001	2.2 (1.4–3.5)	0.001	<0.001	3.5 (1.5–7.9)	0.003	0.003	4.0 (1.2–13.8)	0.027
pN (0 vs. 1-2)	0.030	1.4 (0.8–2.3)	0.19	0.009	1.8 (0.9–3.4)	0.072	0.002	2.8 (1.2–6.3)	0.015
Histology (adenocarcinoma vs. squamous)	0.013	1.4 (0.9–2.2)	0.15	0.067	-	-	0.19	-	-
Naples group (0-1 vs. 2)	0.011	1.3 (0.9–1.9)	0.13	<0.001	2.5 (1.4–4.3)	0.001	0.001	3.5 (1.6–7.9)	0.002

**Table 3 diagnostics-13-03641-t003:** Standardized differences before and after propensity score matching.

	Before Matching				After Matching			
	Naples Group 0-1	Naples Group 2	*p*-Value	Standardized Difference	Naples Group 0-1	Naples Group 2	*p*-Value	Standardized Difference
Gender male, *n* (%)	102 (58.6)	66 (76.7)	0.004	0.39	59 (76.6)	57 (74.0)	0.71	0.06
Age > 72 years, *n* (%)	71 (40.8)	47 (54.7)	0.035	0.28	39 (50.6)	38 (49.4)	0.87	0.02
Smoker (former or current), *n* (%)	143 (82.2)	80 (93.0)	0.019	0.33	71 (92.2)	71 (92.2)	1.00	0.00
Type of resection, *n* (%)			0.13	0.19			0.57	0.09
Sublobar	36 (20.7)	25 (29.1)			17 (22.1)	20 (26.0)		
Major	138 (79.3)	61 (70.9)			60 (77.9)	57 (74.0)		
pT, *n* (%)			0.008	0.36			1.00	0.00
T1	87 (50.0)	28 (32.6)			25 (32.5)	25 (32.5)		
T2-T3-T4	87 (50.0)	58 (67.4)			52 (67.5)	52 (67.5)		
pN, *n* (%)			0.77	0.04			0.52	0.10
N0	141 (81.0)	71 (82.6)			62 (80.5)	65 (84.4)		
N1-N2	33 (19.0)	15 (17.4)			15 (19.5)	12 (15.6)		
Histology, *n* (%)			0.41	0.11			0.86	0.02
Adenocarcinoma	126 (72.4)	58 (67.4)			54 (70.1)	53 (68.8)		
Squamous cell carcinoma	48 (27.6)	28 (32.6)			23 (29.9)	24 (31.2)		

**Table 4 diagnostics-13-03641-t004:** Post-propensity Cox regression analysis. CI: confidence interval; F: female; HR: hazard ratio; M: male.

	Disease-Free Survival			Overall Survival			Cancer-Related Survival		
	Univariable	Multivariable		Univariable	Multivariable		Univariable	Multivariable	
	*p*-Value	HR (95% CI)	*p*-Value	*p*-Value	HR (95% CI)	*p*-Value	*p*-Value	HR (95% CI)	*p*-Value
Age (≤72 vs. >72) years	0.11	-	-	0.78	-	-	0.80	-	-
Gender (M vs. F)	0.42	-	-	0.77	-	-	0.50	-	-
Smoking history (never vs. former/current)	0.59	-	-	0.64	-	-	0.60	-	-
Surgical procedure (major vs. sublobar)	0.013	2.1 (1.2–3.6)	0.006	0.72	-	-	0.76	-	-
Side of surgery (right vs. left)	0.53	-	-	0.81	-	-	0.93	-	-
Lobe (upper and middle vs. lower)	0.68	-	-	0.12	-	-	0.24	-	-
pT (1 vs. 2-3-4)	0.011	2.3 (1.3–4.4)	0.007	0.008	5.2 (1.6–17.0)	0.007	0.046	7.0 (0.9–53.7)	0.061
pN (0 vs. 1-2)	0.31	-	-	0.077	-	-	0.027	2.7 (0.9–7.4)	0.061
Histology (adenocarcinoma vs. squamous)	0.26	-	-	0.99	-	-	0.84	-	-
Naples group (0-1 vs. 2)	0.34	-	-	0.023	2.5 (1.2–5.2)	0.018	0.015	5.2 (1.5–18.2)	0.010

## Data Availability

Data are contained within the article.

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
