# Peer review of "The Naples Prognostic Score Is a Useful Tool to Assess Surgical Treatment in Non-Small Cell Lung Cancer"

_diagnostics, 2023, doi:10.3390/diagnostics13243641_

Round 1

Reviewer 1 Report

Comments and Suggestions for Authors

I read with interest the study entitled 'Naples Prognostic Score Is a Useful Tool to Assess the Surgical Treatment in Non-small Cell Lung Cancer'. The Naples Prognosis Score is an innovative prognostic biomarker-based tool for evaluating immune and nutritional status and can indicate prognosis.  The paper presents a thorough analysis of the NPS as a predictor of long-term survival rates amongst NSCLC patients following surgical treatment. My plagiarism check result is %18. In summary, the paper presents a thorough analysis of the NPS as a predictor of long-term survival rates amongst NSCLC patients following surgical treatment. The study's stated aim is to examine the correlation between NPS and long-term survival among NSCLC patients subjected to surgical intervention. The study methodology, particularly the retrospective propensity score matching analysis, is adequately described. The article presents comprehensive demographic and clinical characteristics of the enrolled population, which enhances the analysis's depth. The statistical methods employed, such as Kaplan-Meier curves, Cox regression analysis, ROC curves, and propensity score matching, boost the study's reliability. The patient inclusion and exclusion criteria have been clearly outlined, thereby enhancing the study's methodological validity. The discussion section could benefit from a more critical analysis of the results within the context of existing literature. Further elaboration on the limitations and potential sources of bias is necessary. Although these limitations have been acknowledged, a detailed discussion on how they may affect the study's findings and generalizability would bolster the article's credibility. The conclusion presents a robust summary of the findings and highlights the possible clinical ramifications. However, reiterating the limitations and the requirement for further validation studies may enhance the conclusion. The article has a sturdy basis and offers valuable insights into the potential prognostic relevance of NPS in NSCLC patients post operation. Implementing the recommended enhancements could enhance the lucidity and effect of the discoveries. The availability of comparable studies in the literature will enhance comprehensive systematic reviews, meta-analyses, and self-sufficient outcomes.

Comments on the Quality of English Language

There are some minor grammatical and syntactical errors throughout the article which would benefit from a thorough proofread. 

Author Response

Dear Reviewers,

Thank you very much indeed for reviewing our manuscript titled ‘’Naples Prognostic Score is a useful tool to assess the surgical treatment in Non-Small Cell Lung Cancer’’. We really appreciate the time you invested in it and your valuable comments resulting in improving it.

We have performed all changes as suggested below (please see our responses in red and the changes or additions highlighted in the text) and we do hope our manuscript meets now the journal’s requirements.

Should you need anything further please do not hesitate to contact me.

Kind regards,

Prof Stefano Elia

Review 1

I read with interest the study entitled 'Naples Prognostic Score Is a Useful Tool to Assess the Surgical Treatment in Non-small Cell Lung Cancer'. The Naples Prognosis Score is an innovative prognostic biomarker-based tool for evaluating immune and nutritional status and can indicate prognosis.  The paper presents a thorough analysis of the NPS as a predictor of long-term survival rates amongst NSCLC patients following surgical treatment.

My plagiarism check result is %18. 

In summary, the paper presents a thorough analysis of the NPS as a predictor of long-term survival rates amongst NSCLC patients following surgical treatment. The study's stated aim is to examine the correlation between NPS and long-term survival among NSCLC patients subjected to surgical intervention. The study methodology, particularly the retrospective propensity score matching analysis, is adequately described. The article presents comprehensive demographic and clinical characteristics of the enrolled population, which enhances the analysis's depth. The statistical methods employed, such as Kaplan-Meier curves, Cox regression analysis, ROC curves, and propensity score matching, boost the study's reliability. The patient inclusion and exclusion criteria have been clearly outlined, thereby enhancing the study's methodological validity. Thank you.

 The discussion section could benefit from a more critical analysis of the results within the context of existing literature. Further elaboration on the limitations and potential sources of bias is necessary. Although these limitations have been acknowledged, a detailed discussion on how they may affect the study's findings and generalizability would bolster the article's credibility.

Thank you. We have addressed the matter and modified the discussion as per your suggestions.

The conclusion presents a robust summary of the findings and highlights the possible clinical ramifications. Thank you

However, reiterating the limitations and the requirement for further validation studies may enhance the conclusion. Thank you. We have followed your suggestion.

The article has a sturdy basis and offers valuable insights into the potential prognostic relevance of NPS in NSCLC patients post operation. Thank you

Implementing the recommended enhancements could enhance the lucidity and effect of the discoveries. The availability of comparable studies in the literature will enhance comprehensive systematic reviews, meta-analyses, and self-sufficient outcomes.

Recommended improvements have been completed as above. A thorough proofread has been performed to correct minor grammatical and syntactical errors.

Reviewer 2 Report

Comments and Suggestions for Authors

Dear authors, you performed  a retrospective analysis to determine if the Naples score is corralated with recurrence and overall survival after lung cancer resection. You reported that the Naples score is a significant prognostic factor. Some remarks:

- It is unclear for me the rate of wedge for NSCLC (almost 20%). Can you explain why? compromised patients? It should be mentioned in limitations and explained in discussion.

- There are no informations on comorbidities (cardio-pulmonary or pulmonary functions). The propensity score included only age and type of surgery. This is an important factor that can explain the Naples score and survival. This is the main limitation of your study and should be included in the discussion. It would have been interesting to include comorbidities index (like Charslon Score), but of course due to retrospective nature of the szudy, it should be difficult.

Author Response

Review 2

Dear authors, you performed  a retrospective analysis to determine if the Naples score is corralated with recurrence and overall survival after lung cancer resection. You reported that the Naples score is a significant prognostic factor. Some remarks:

- It is unclear for me the rate of wedge for NSCLC (almost 20%). Can you explain why? compromised patients? It should be mentioned in limitations and explained in discussion.

Many thanks for your comment. This has been addressed now and a relevant addition has been made in the results section and subsequently to the discussion.

- There are no informations on comorbidities (cardio-pulmonary or pulmonary functions). The propensity score included only age and type of surgery. This is an important factor that can explain the Naples score and survival. This is the main limitation of your study and should be included in the discussion. It would have been interesting to include comorbidities index (like Charlson Score), but of course due to retrospective nature of the study, it should be difficult.

Thank you, this has been addressed now. Indeed, Charlson would be difficult to include at this point due to the retrospective nature of the study, many thanks for considering this.